# Effect of Combining the Ionophore Monensin with Natural Antimicrobials Supplemented in the Last Phase of Finishing of Lambs: Growth Performance, Dietary Energetics, and Carcass Characteristics

**DOI:** 10.3390/ani13162547

**Published:** 2023-08-08

**Authors:** Alfredo Estrada-Angulo, Lucía de G. Escobedo-Gallegos, Yesica J. Arteaga-Wences, Jorge L. Ramos-Méndez, Jesús A. Quezada-Rubio, Claudia A. Vizcarra-Chávez, Yissel S. Valdés-García, Alberto Barreras, Richard A. Zinn, Alejandro Plascencia

**Affiliations:** 1Faculty of Veterinary Medicine and Zootechnics, Autonomous University of Sinaloa, Culiacan 80260, Mexico; alfred_vet@hotmail.com (A.E.-A.); lucia.escobedo@uabc.edu.mx (L.d.G.E.-G.); arteaga.yesi.92@hotmail.com (Y.J.A.-W.); ramos.jorge.92@outlook.com (J.L.R.-M.); jesus.quezada.fmvz18@uas.edu.mx (J.A.Q.-R.); claudia.vizcarra2208@gmail.com (C.A.V.-C.); 2Veterinary Science Research Institute, Autonomous University of Baja California, Mexicali 21100, Mexico; yissel.valdes@uabc.edu.mx (Y.S.V.-G.); beto_barreras@yahoo.com (A.B.); 3Animal Science Department, University of California, Davis, CA 95616, USA; razinn@ucdavis.edu

**Keywords:** natural additives, monensin, lambs, growth-performance, energetics, carcass

## Abstract

**Simple Summary:**

The ionophore monensin (MON) is an antibiotic widely used in several countries to increase feed efficiency and the health of feedlot cattle. Due to the fact that consumers increasingly claim meat products are free of antibiotics, it is necessary to search for alternatives to reduce, or even avoid, the antibiotics used as growth promoters in feedlots. Research into the evaluation of the effects of combining MON with natural feed additives on productivity is a step towards identifying ways to reduce, or even replace the use of MON in feedlots. This study reveals that lambs showed better efficiencies when MON is combined with probiotics but not when MON is combined with essential oils; moreover, probiotics supplemented alone can fully replace MON without detriment to efficiency or carcass characteristics.

**Abstract:**

With the aim of evaluating the effect of combining an antibiotic ionophore with plant extracts and probiotics on the productive efficiency (performance and carcass) during the last phase of lamb fattening, 24 Pelibuey × Katahdin male lambs (38.47 ± 3.92 kg, initial weight) were fed with a high-energy diet during for 56 days, and assigned, under a complete randomized block design experiment to one of the following supplement treatments: (1) 28 mg of monensin/kg diet DM supplemented alone (MON), (2) combination of MON plus 2 g/kg diet of a product contained *Bacillus subtilis* 2.2 × 10^8^ CFU kg diet DM (MON + BS), (3) combination of MON + BS plus 300 mg essential oils/kg diet DM (MON + BS + EO), and (4) BS alone. At the end of the feeding trial (56-d), lambs were slaughtered and carcass variables were measured. Compared to the rest of the treatments, combining MON with BS improved dietary NE by 3.4% and the efficiency of utilization of dietary energy consumed. Inclusion of EO in the MON + BS combination resulted in a similar average daily weight gain (ADG) and feed efficiency (GF) when compared with MON + BS, but showed a lower dietary net energy (NE), hot carcass weight, and dressing percentage. Lambs receiving BS alone showed greater average ADG and dry matter intake (DMI) than lambs receiving MON + BS + EO, but similar feed GF and dietary NE. There were no treatment effects on tissue composition, whole cut, or visceral organ mass. It was concluded that combining probiotics with the ionophore monensin can improve the efficiency of dietary energy utilization in the last phase of finishing. Probiotics supplemented alone result in greater ADG without a difference in dietary energy efficiency when compared with MON alone. Inclusion of EO in the MON + BS combination did not show advantages; on the contrary, it reduced carcass weight and dressing percentage. It is necessary to further research the potential complementary effects of combining diverse sources of natural additives with synthetic antibiotics.

## 1. Introduction

Due to the changes in gain composition observed during the finishing phase when lambs are fed high-energy diets, there is less efficiency in the use of dietary energy for growth [1]. In addition, during the finishing phase, feedlot lambs consume diets containing large amounts of soluble carbohydrates. This feeding system represents a high risk for the presence of ruminal subacute acidosis episodes, which can negatively affect feed efficiency at this stage [2]. Improved ruminal fermentation by modulating certain microorganisms that reduce production of specific intermediate organic acids (i.e., lactate), as well as promoting an increase in ruminal propionate by reducing ruminal acetate and enhancing epithelial health and increasing nutrient absorption, is one of the ways to obtain better performance in ruminants that are fed high-energy diets [3]. A widely used tool to prevent this type of disorder and increase efficiency in the final stage of fattening is the antibiotic monensin (MON), which has been used for many years as a growth promoter in feedlots in several countries (i.e., Mexico, Canada, USA, Brazil, New Zealand, Argentina, Chile, and South Africa, among others) where its use is approved [4,5,6]. However, consumers increasingly claim that meat products are free of antibiotics. This concern has led livestock industries to look for possible alternatives, among which are, natural additives such as probiotics and essential oils [7]. Probiotics and essential oils alter rumen fermentation and promote gastrointestinal (GIT) health; in addition, some antioxidant and immunological effects have been attributed; all these effects aid several beneficial responses primarily on growth rate and/or feed efficiency in lambs [8,9,10,11]. Therefore, probiotics and essential oils favor changes in ruminal microorganisms and gastrointestinal health analogous to antibiotics [12]. Nonetheless, although the mechanism of action of probiotics and essential oils is not fully understood, they appear to act through different routes than antibiotics [13,14]. Thus, a combination of probiotics and essential oils with synthetic antibiotics could have complementary effects. Nevertheless, most of the studies have been directed at evaluating productivity efficiency when using single additives, but few reports are available about the effects of supplementing with this type of combination. Generating this type of information could promote future studies about decreasing doses of antibiotics used as feed additives through combinations with natural additives. Furthermore, the strategy to combine antibiotics with natural additives could start the transition from the use of antibiotics as growth promoters to the use of natural alternatives in the feedlot industry. At present, there is little information available regarding the effects of combining MON with natural additives. For this reason, the objective of this experiment was to evaluate the effect of combining MON with probiotics, and with probiotics plus essential oils, on growth performance, dietary energetics, carcass characteristics, and visceral mass in lambs finished with high-energy diets. In addition, we include a comparison between MON and a probiotic (*Bacillus subtilis*) when both are supplemented alone. 

## 2. Materials and Methods

The experiment was conducted at the Universidad Autónoma de Sinaloa Feedlot Lamb Research Unit, located in Culiacán City, México (24°46′13″ N and 107°21′14″ W). Culiacán City is about 55 m above sea level and has a tropical climate. During the course of the experiment, the ambient air temperature averaged 22.6 °C (minimum and maximum of 19.4 °C and 25.7 °C, respectively), and relative humidity averaged 54.0% (minimum and maximum of 53.8% and 66.3%, respectively). All animal management procedures were conducted within the guidelines of federally and locally-approved techniques for animal use and care [15] and approved by the Ethics Committee of the Faculty of Veterinary Medicine and Zootechnics from the Autonomous University of Sinaloa (Protocol #09162022).

### 2.1. Animals, Experimental Design, and Diets

Thirty Pelibuey × Katahdin crossbred intact male lambs were received at the research facility 40 days before the start of the trial. Upon arrival, lambs were treated for parasites (7.5 mg/kg LW; Closantel Panavet 15%, Panamericana Veterinaria de México City, México), injected with 2 mL vitamin A (500,000 UI, 75,000 IU vitamin D3, and 50 IU vitamin E; Synt-ADE^®^, Zoetis México, México City, México), and vaccinated for Mannheimia haemolytica (One Shot Ultra, Zoetis México, México City, México). For 3 weeks before the initiation of the experiment, lambs were fed the basal diet used during the experimental period. Following a 2-week evaluation period, lambs were individually weighed before the morning meal (electronic scale; TORREY TIL/S: 107 2691, TOR REY Electronics Inc., Houston, TX, USA). From the original group of 30 lambs, 24 lambs (38.47 ± 3.92 kg, initial weight, BW) were selected, based on the uniformity of weight and general condition, for use in the experiment and were assigned to one of four weight groupings (blocks) in 24 pens (one lambs/pen and 6 replicas per treatment). Pens have 6 m^2^ with overhead shade, automatic waterers, and 1 m fence-line feed bunks. A cracked corn-based total mixed ration was used as a basal diet (white corn cracked for a final density of approximately 0.52 kg/L) in which ground sudangrass hay was used as a forage source. Sudangrass hay was grounded in a hammer mill (Azteca 20, Molinos Azteca, Guadalajara, México) with a 3.81-cm screen before incorporation into the total mixed ration. The ingredients and chemical composition of the basal diet are shown in Table 1. Treatments consisted of a basal diet supplemented with: (1) 28 mg of monensin/kg diet DM (MON), (2) a combination of MON plus 2 g/kg diet of a product containing *Bacillus subtilis* 2.2 × 10^8^ CFU kg diet DM (MON + BS), (3) a combination of MON + BS plus 300 mg essential oils/kg diet DM (MON + BS +EO), and (4) 2 g/kg diet of a product containing *Bacillus subtilis* 2.2 × 10^8^ CFU kg diet (BS). The source of ionophore monensin used was Rumensin 90 (Elanco Animal Health, Indianapolis, IN, USA). The source essential oils used were PrintArome (NOREL Nutrición Animal, Querétaro, México), a blend of thymol, carvacrol, and cinnamaldehyde. Finally, the source of *Bacillus subtilis* was CLOSTAT dry (Kemin Industries, Des Moines, IA, USA), which contained 2.2 × 10^8^ CFU of *Bacillus subtilis*. The doses used for Rumensin, PrintArome, and CLOSTAT followed the recommendations expressed in the Fact Sheet for each additive. The treatments (complete mixed diets) were prepared using a 2.5 m^3^ capacity paddle mixer (model 30910-7, Coyoacán, México). To avoid contamination between treatments, the mixer was thoroughly cleaned between each elaborate batch. To ensure additive consumption, the total daily dosage per lamb was mixed into 300 g of basal diet provided in the morning feeding (all lambs were fed the basal control diet in the afternoon feeding).

### 2.2. Measurements and Samplings

Lambs were provided fresh feed twice daily, at 08:00 and 14:00 h. Whereas the amount of feed provided in the morning feeding was constant, the feed offered in the afternoon feeding was adjusted daily, allowing for a feed residual of approximately 50 g/kg per day. The residual feed of each pen was collected between 07:40 and 07:50 h each morning, composited through the experiment, and weighed at the end of the experiment to determine the feed intake. The adjustments to either increase or decrease daily feed delivery were provided in the afternoon feeding. Lambs were weighed just prior to the morning feeding on day 1 and at the conclusion of the experiment. Live weights (LW) on day 1 was converted to shrunk body weight (SBW) by multiplying LW by 0.96 to adjust for the gastrointestinal fill [17]. All lambs were fasted (for feed but not for drinking water) for 18 h before recording the final LW.

Feed samples were collected for each elaborate batch. Feed refusal was collected daily and composited weekly for DM analysis (oven drying at 105 °C until no further weight loss; method 930.15) [18]. 

### 2.3. Chemical Analysis

Feed samples were subjected to the following analyses: DM (oven drying at 105 °C until no further weight loss; method 930.15) and CP (N × 6.25, method 984.13) according to AOAC [18]. Neutral detergent fiber (NDF) was determined following procedures described by Van Soest et al. (corrected for NDF-ash, incorporating heat-stable α-amylase using Ankom Technology, Macedon, NY, USA) [19].

### 2.4. Calculations

Estimates of ADG and dietary net energy are based on initial SBW and final (d 56) fasted SBW. The average daily gain was computed by subtracting the initial SBW from the final SBW and dividing the result by the number of days on feed. Feed efficiency was computed as ADG/daily DMI. One approach for evaluating the efficiency of dietary energy utilization in growth-performance trials is the ratio of observed-to-expected DMI and observed-to-expected dietary NE. Based on estimated diet NE concentrations and measures of growth performance, there is an expected energy intake. This estimation of expected DMI is performed based on the observed ADG, average SBW, and NE values of the diet (Table 1). The expected DMI, kg/d = (EM/2.03) + (EG/1.39), where EM (energy required for maintenance, Mcal/d) = 0.056 × SBW^0.75^, EG (energy gain, Mcal/d) = 0.276 × ADG × SBW^0.75^, and 2.03 and 1.39 are the NE_m_ and NE_g_ values contained in the basal diet; those values were calculated based on the ingredient composition [16] in the basal diet (Table 1). The coefficient (0.276) was taken from NRC [20], assuming a mature weight of 113 kg for Pelibuey × Katahdin male lambs [21]. The observed dietary net energy for maintenance was calculated using EM and EG values and the DMI observed during the experiment by means of the quadratic formula: $$x=\frac{-b\pm \sqrt{b^{2}-4ac}}{2c}$$
where: x = NE_m_, Mcal/kg, a = −0.41EM, b = 0.877 EM + 0.41 DMI + EG, and c = −0.877 DMI [22]. Dietary NE for gain was derived assuming that NE_g_ = 0.877NE_m_ − 0.41 [22].

### 2.5. Carcass Characteristics, Whole Cuts, and Tissue Shoulder Composition

All lambs were harvested on the same day. After humanitarian sacrifice, lambs were skinned, and the gastrointestinal organs were separated and weighed. After carcasses (with kidneys and internal fat included) were chilled in a cooler at −2 to 1 °C for 24 h, the following measurements were obtained: (1) body wall thickness (at a point between the 12th and 13th rib, five inches from the midline of the carcass); (2) fat thickness perpendicular to the m. longissimus thoracis (LM), measured over the center of the ribeye between the 12th and 13th rib; (3) LM surface area, measured using a grid reading of the cross sectional area of the ribeye between the 12th and 13th rib; and (4) kidney, pelvic, and heart fat (KPH). The KPH was manually removed from the carcass, weighed, and reported as a percentage of the cold carcass weight [23]. Each carcass was split into two halves. The left side was fabricated into wholesale cuts without trimming, according to the North American Meat Processors Association guidelines [24]. Rack, breast, shoulder, and foreshank were obtained from the foresaddle, and the loins, flank, and leg from the hindsaddle. The weight of each cut was subsequently recorded. The tissue composition of the shoulder was assessed using physical dissection by the procedure described by Luaces et al. [25].

### 2.6. Visceral Mass Data

Components of the digestive tract (GIT), including the stomach complex (rumen, reticulum, omasum, and abomasum), liver, small intestine (duodenum, jejunum, and ileum), and large intestine (caecum, colon, and rectum), were removed and weighed. The GIT was then washed, drained, and weighed to obtain empty weights. The difference between full and washed digesta-free GIT was subtracted from the final BW to determine empty body weight (EBW). All tissue weights are reported on a fresh tissue basis. Organ mass is expressed as grams of fresh tissue per kilogram of final EBW, where final EBW represents the final full live weight minus the total digestive weight. The stomach complex was calculated as the digesta-free sum of the weights of the rumen, reticulum, omasum, and abomasum. 

### 2.7. Statistical Analysis

All data (gain, gain efficiency, and dietary energetics, DM intake, carcass, and visceral mass) were analyzed as a randomized complete block design, of which the initial weight was the blocking criterion and lamb was considered the experimental unit using the MIXED procedure of SAS software V9.1 [26], where fixed effects were treatment and block, and lamb into treatment as the random component. Dietary treatments were randomly assigned to lambs within blocks, with six replicas per treatment according to the following statistical model: Yij = µ + Bi + Tj + εij, where, µ is the common experimental effect, B_i_ represents the initial weight block effect, T_j_ represents the dietary treatment effect, and ε_ij_ represents the residual error. Treatment effects were separated by using orthogonal contrasts. In all cases, the least squares mean and standard error are reported, and contrasts are considered significant when the *p* value ≤ 0.05.

## 3. Results and Discussion

No morbidity or mortality was observed during the experimental period. The effects of treatments on growth performance and dietary energy are shown in Table 2. Based on intake and doses used of each additive, the daily average net intake of additives (alone and combined) per lamb were 32 mg for MON, 2.4 g for BS, and 341 mg for EO, respectively. Daily intake of MON was within the range of 28 to 42 mg/d, which consistently increased feed efficiency, or dietary net energy, in finishing lambs [27,28,29]. However, the magnitude of response to MON supplementation can vary from nil to 4% in lambs that have been fed diets containing dietary NE higher than 2.00 Mcal/kg [29,30,31]. The basis of this is not completely understood, but a high soluble carbohydrates-to-NDF ratio in diet (i.e., >4) and environmental factors (high ambient heat load) could be the main factors that can affect the magnitude of the MON response regarding the improvement in efficiency of utilization of diet energy [32,33]. Because the present experiment was performed under favorable environmental conditions, the proportion of CHOS: NDF in diet could be the factor that explains the low increase (1%) of the observed over-expected dietary NE for lambs receiving MON. Although soluble CHO was not determined in diets, according to the NRC [16], the estimated soluble CHO in diets was 64%, representing a high soluble CHO to NDF ratio of 4.3. All treatments that include MON (alone or combined) showed a lower (11.9%, *p* ≤ 0.05) DMI compared to BS supplemented alone. A decrease in DMI caused by MON supplementation is a common response [32], but contrary to our expectations, this effect persisted even when it was combined with BS and with BS+EO. The absence of an effect on DMI from BS when combined with MON is uncertain. Increases in DMI, and in turn in ADG, by the inclusion of probiotics in the diets have been reported in several reports [34,35]. Although the magnitude of response to DMI and ADG can vary by type and dose of probiotic used, associative effects with diet composition, and ambient heat load [36,37].

Lambs receiving the BS alone showed 19.0% (*p* = 0.04) and 14.3% greater (*p* = 0.02) average ADG and dry matter intake (DMI) than lambs receiving MON + BS + EO, but similar (*p* ≥ 0.15) feed GF and dietary NE. Similarly, BS supplemented alone showed greater DMI (11%, *p* = 0.05) but similar (*p* ≥ 0.13) feed efficiency and dietary energy to MON. The strain of *Bacillus subtilis* has not been thoroughly investigated in lambs. In poultry, BS only increased feed efficiency when it was supplemented under heat-stress conditions [38]. Similarly, milking cows in heat-stress environments that were supplemented with BS showed better milk yield than those that were not supplemented [39]. Supplementation with a daily dose of 2.56 × 10^9^ viable spores of BS resulted in increases in DMI and milk yield in goats [40]. Although there is no information about the effect of BS supplementation in lambs fed with a high-energy diet, increases in DMI and ADG in lambs supplemented with BS have been reported previously when they were fed a moderate-energy diet [40,41].

According to previous reports [42,43], the combination of MON with BS did not modify DMI or ADG compared to the rest of the treatments. However, as observed in the current study, this combination increased 3.4% the net energy utilization of feed (1.045 vs. 1.009) when offered in a short-term period (45 d) in feedlot cattle [42]. As was previously exposed [44], the estimation of dietary NE based on measures of growth performance provides important insight into potential additive (or other factors) effects on the efficiency of dietary energy utilization. An observed-to-expected dietary NE ratio of 1.00 indicates that performance is consistent with dietary NE values based on tables of feedstuff standards [16] and observed DMI. A ratio that is greater than 1.00 is indicative of greater efficiency in dietary energy utilization. Whereas, a ratio that is lower than 1.00 indicates lower than expected efficiency of energy utilization. Therefore, the combination of MON and BS increased the efficiency of dietary NE utilization compared to the other treatments. As mentioned previously, probiotics favor changes in ruminal microorganisms and gastrointestinal health analogous to antibiotics. An increase in the ability to obtain energy from feeds, improving energy efficiency, is explained by the shift to a ruminal microbiome that is less complex but more specialized to support the host’s energy needs [45]. Those authors indicate that increases in ruminal propionate with decreases in ruminal acetate and methane production, as happens when probiotics or ionophores are supplemented, are one of the main explanations for the increased efficiency of diet energy utilization. Another factor that can help improve the efficiency of dietary energy utilization is its anti-oxidative stress properties. It has been demonstrated that supplemental *Bacillus subtilis* decreased cellular oxidative stress in lambs [8]. Furthermore, it is well known that the quantity of metabolizable amino acids that reach the intestine can affect the efficiency of dietary energy utilization in ruminants [46,47]. Several studies have demonstrated that MON decreases ruminal microbial protein (MP) synthesis, affecting the quantity of MP that flows to the intestine [48,49,50,51]. On the other hand, probiotics consistently increased ruminal MP synthesis [34,52,53,54]. Thus, improvements in dietary energy in lambs that received MON + BS could be due to the fact that BS promotes greater MP synthesis, which is negatively affected by MON, in such a manner that it is possible that BS has a complementary effect, alleviating in this manner the negative effect of MON in MP, promoting greater quantities of MP reaching the duodenum, and increasing in this way the dietary energy utilization, which indicates greater than expected efficiency of energy utilization.

Including EO in the MON + BS combination did not improve responses or performance when compared with MON alone or the MON + BS combination. In contrast, lambs receiving MON + BS + EO showed a numerically 10.3% (*p* = 0.12) lower ADG than lambs receiving MON + BS treatment. Moreover, compared to MON + BS, the inclusion of EO decreased hot carcass weight and dressing percentage. The negative effects of EO inclusion in the MON + BS combination could be explained by possible antagonism between EO and MON instead of possible antagonism between BS and EO. This can be inferred from the fact that although there is very little information regarding the combination of probiotics with EO in ruminants, a report indicated that the combination of *Lactobacillus acidophilus* plus *S. cerevisiae* with a blend of EO improved daily gain and feed efficiency in calves, showing a synergistic beneficial effect [55].

On the other hand, no information is available with respect to the effects of a specific combination of BS and EO on growth performance in ruminants, but in breeder broilers, the combination improved reproductive performance [56]. Similarly, when 1 g of *Bacillus subtilis* was combined with 300 mg of a blend of essential oils composed of thymol plus carvacrol (additive/kg of diet) and offered to weaned pigs rearing under high ambient heat, supplemented pigs showed an increase in average daily gain [57]. Therefore, it can be supposed that the combination of probiotics with EO did not have an antagonistic effect on the performance of lambs in this experiment. In contrast, the combination of MON with EO in ruminant diets has shown no complementary effects; even more, some results indicate antagonism effects on growth performance and digestibility. At the time of writing this report, there was no information regarding the effects of combination of MON + EO on growth performance and carcasses of feedlot lambs, but information about the topic is available on feedlot cattle and dairy cows. In an earlier report, Benchaar et al. [58] concluded that there were no interaction effects with the combination MON + EO (350 mg/day and 2000 mg/day, respectively) on productivity and ruminal fermentation in lactating dairy cows. More recently [59], an experiment was performed to evaluate the effects of supplemental MON (200 mg/day) alone or combined with EO (1500 mg/day) on angus steers consuming a silage-based diet. Steers that were fed with the MON + EO combination had lower ADG and feed efficiency than steers that received MON alone. In the same way, a study conducted with yearling Holstein bulls by Wu et al. [60] tested supplementation with MON (25 mg/kg diet), EO (mainly composed of carvacrol and thymol, 26 mg/kg diet), or a combination of MON and EO. The diet used was a growing diet based on yellow corn silage. The experiment lasted 310 days. Bulls that were fed with supplemental MON and EO registered greater ADG and feed efficiency (1.19 and 1.22 kg) when compared to the MON + EO combination (1.11 kg/d). Those researchers indicate that antagonism could be caused by the similarity of modes of action on microbes between MON and EO. Potential dual-action modes that occur simultaneously when feeding MON and EO at the same time may result in negative effects that are not present when fed separately [60]. This is in line with the report of Latack et al. [61], which involves a growth trial and a digestion trial. Steers were fed a high-energy, steam-flaked corn-based diet. Steers that received the combination of MON + EO showed a lower final weight ADG than MON alone. Moreover, the combination tended (*p* = 0.09) to reduce dietary energy and dietary energy efficiency by 5%. The negative effects on performance due to the combination of MON + EO observed in the growth trial were in agreement with decreases in ruminal digestion of organic matter, neutral detergent fiber, and reduction of MN entering the intestine observed in the digestion trial for the MON + EO treatment. The previous results and the results obtained in the experiment presented here regarding the inclusion of EO in the MON combination confirm that MON plus EO is not an alternative to improving the expected response when additives are supplemented alone.

Treatment effects on carcass and tissue composition characteristics and shoulder tissue composition are shown in Table 3.

The absence of effects on carcass traits and tissue composition of lambs is the most common response to MON [29,62,63,64] and probiotics [37,65,66] when supplemented alone. However, few studies report effects on hot carcass weight and LM area when lambs are supplemented with MON alone [67,68] or probiotics alone [69,70]. These effects are mainly observed when the rate of gain is markedly higher (i.e., >15%) for supplemented groups. It is well known that the main factor that affects HCW and LM is final weight, which is directly related to the rate of gain [67,71]. Compared to MON alone and the combination of MON + BS + EO, the combination of MON + BS increased HCW and dressing percentage without effects on the rest of the variables evaluated. Information about the effects of the combination of MON plus probiotic on carcasses is scant. In light of this, an experiment was carried out in which the combination of *S. cerevisiae* (0.22 g/kg diet) in combination with the ionophore lasalocid (15 mg/kg diet) was evaluated in Mutton sheep fed for 47 days with a high-energy diet. In contrast with our results, carcass traits did not differ when compared to supplementation alone with additives [72]. It has been determined by several studies that the magnitude of the effects of ionophores is dose-dependent [73]. Maximal response in finishing feedlot lambs and cattle has been noted with doses around 30 mg lasalocid/kg diet [33,74]. Thus, it appears that the dose used for the ionophore lasalocid in the experiment of Piennar et al. [72] was half that recommended for maximal response. In such a way that this could be the reason for the absence of an effect in that experiment.

Consistent with previous reports [68,69,73,74], comparing additives either alone or in combination did not affect the whole cut (as % of CCW, Table 4), nor visceral organ mass (as g/kg empty BW, Table 5).

## 4. Conclusions

Combining probiotics with the ionophore monensin can improve the efficiency of dietary energy utilization in the last phase of finishing. Probiotic supplementation alone results in greater ADG without a difference in dietary energy efficiency when compared with MON alone. Inclusion of EO in the MON + BS combination did not show advantages; on the contrary, it reduced carcass weight and dressing percentage, indicating that combinations of MON and EO can have a potential antagonistic effect. It is necessary to further research the potential complementary effects of combining diverse sources of natural additives with synthetic antibiotics.

## Figures and Tables

**Table 1 animals-13-02547-t001:** Composition of dietary treatments offered to the lambs.

	Treatments ^§^			
Item	MON	MON + BS	MON + BS + EO	BS
Ingredient composition, % DM basis				
Dry-rolled corn	55.00	55.00	55.00	55.00
Sudangrass hay	10.50	10.50	10.50	10.50
Soybean meal	15.00	15.00	15.00	15.00
Monensin	+++	+++	+++	0.00
*Bacillus subtillis*	0.00	+++	+++	+++
Essential oils	0.00	0.00	+++	0.00
Molasses cane	10.00	10.00	10.00	10.00
Zeolite	2.50	2.50	2.50	2.50
Yellow grease	4.00	4.00	4.00	4.00
Mineral-protein supplement *	2.50	2.50	2.50	2.50
Chemical composition (%DM basis) ^‡^				
Dry matter	88.82	88.68	88.70	88.77
Neutral detergent fiber	15.05	15.05	15.05	15.05
Crude protein	15.43	15.43	15.46	15.43
Ether extract	6.60	6.60	6.60	6.60
Calculated net energy (Mcal/kg) ^†^				
Maintenance	2.03	2.03	2.03	2.03
Gain	1.39	1.39	1.39	1.39

The symbols “+++” denote additive inclusion; ^§^ MON = Monensin 28 mg/kg diet DM (Rumensin 90^®^, Elanco Animal Health, Indianapolis, IN, USA); MON + BS = combination of MON plus 2 g/kg diet of a product contained *Bacillus subtilis* 2.2 × 10^8^ CFU (CLOSTAT dry, Kemin Industries, Des Moines, IA, USA); MON + BS+ EO = combination of MON + BS plus 300 mg essential oils (PrintArome, NOREL Nutritición Animal, Queretaro, Mexico); BS = 2 g/kg diet of a product contained *Bacillus subtilis* 2.2 × 10^8^ CFU (CLOSTAT dry, Kemin Industries, Des Moines, IA, USA). * Mineral premix contained: CP, 53.0%; calcium, 13.6%; phosphorus, 0.40%; magnesium, 1.0%; potassium, 0.71%; NaCl, 15%, CoSO_4_, 0.01%; CuSO_4_, 0.14%; FeSO_4_, 0.47%; ZnO, 0.16%; MnSO_4_, 0.14%; KI, 0.008%. ^‡^ Based on tabular values for individual feed ingredients [16], with the exception of CP and NDF, which were determined in our laboratory. ^†^ Based on tabular energy values for individual feed ingredients informed by the NRC [16].

**Table 2 animals-13-02547-t002:** Effect of treatments on growth performance of finishing lambs.

	Treatments ^†^					
Item	MON	MON + BS	MON + BS + EO	BS	SEM	*p*-Value
Days on the test	56	56	56	56		
Pen replicates	6	6	6	6		
Live weight, kg/d						
Initial	38.64	38.53	38.49	37.99	0.499	0.80
Final	50.34 ab	50.95 ab	49.63 a	51.77 b	0.787	0.15
Average daily gain, kg/d	0.208 ab	0.222 ab	0.199 a	0.246 b	0.015	0.13
Dry matter intake, kg/d	1.181 a	1.188 a	1.137 a	1.327 b	0.038	0.04
Gain to feed ratio, kg/kg	0.177	0.186	0.175	0.186	0.005	0.22
Diet net energy, Mcal/kg						
Maintenance	2.055 a	2.120 b	2.060 a	2.043 a	0.011	<0.01
Gain	1.390 a	1.449 b	1.397 a	1.382 a	0.014	<0.01
Observed-to-expected diet NE						
Maintenance	1.011 a	1.045 b	1.015 a	1.001 a	0.005	<0.01
Gain	1.000 a	1.042 b	1.005 a	0.994 a	0.007	<0.01
Observed-to-expected DMI	0.995 a	0.958 b	0.990 a	1.001 a	0.006	<0.01

a,b Means a row with different superscripts differ (*p* < 0.05). ^†^ MON = Monensin 28 mg/kg diet DM (Rumensin 90^®^, Elanco Animal Health, Indianapolis, IN); MON + BS = combination of MON plus 2 g/kg diet of a product contained *Bacillus subtilis* 2.2 × 10^8^ CFU (CLOSTAT dry, Kemin Industries, Des Moines, IA, USA); MON + BS + EO = combination of MON + BS plus 300 mg essential oils (PrintArome, NOREL Nutritición Animal, Queretaro, Mexico); BS = 2 g/kg diet of a product contained *Bacillus subtilis* 2.2 × 10^8^ CFU (CLOSTAT dry, Kemin Industries, Des Moines, IA, USA).

**Table 3 animals-13-02547-t003:** Effect of treatments on carcass characteristics of finishing lambs.

	Treatments ^†^					
Item	MON	MON + BS	MON + BS + EO	BS	SEM	*p*-Value
Hot carcass weight, kg	28.80 ab	30.41 b	28.61 a	30.21 ab	0.522	0.09
Dressing percentage	57.17 a	59.71 b	57.64 a	58.35 ab	0.616	0.06
Cold carcass weight, kg	28.14 a	29.90 b	28.13 a	29.64 ab	0.542	0.07
LM area, cm^2^	17.40	19.02	17.97	17.90	0.951	0.67
Fat thickness, cm ^§^	0.270	0.284	0.300	0.355	0.024	0.81
Kidney, pelvic, and heart fat, %	3.45	3.68	3.00	3.55	0.247	0.29
Carcass yield *	1.45	1.52	1.58	1.47	0.096	0.83
Shoulder composition, %						
Muscle	70.10	69.74	69.65	69.34	0.866	0.09
Fat	13.00	13.97	14.23	14.44	1.079	0.12
Muscle to fat ratio	5.39	4.99	4.89	4.80	0.400	0.12

a,b Means a row with different superscripts differs (*p* < 0.05). ^†^ MON = Monensin 28 mg/kg diet DM (Rumensin 90^®^, Elanco Animal Health, Indianapolis, IN); MON + BS = combination of MON plus 2 g/kg diet of a product contained *Bacillus subtilis* 2.2 × 10^8^ CFU (CLOSTAT dry, Kemin Industries, Des Moines, IA, USA); MON + BS + EO = combination of MON + BS plus 300 mg essential oils (PrintArome, NOREL Nutritición Animal, Queretaro, Mexico); BS = 2 g/kg diet of a product contained *Bacillus subtilis* 2.2 × 10^8^ CFU (CLOSTAT dry, Kemin Industries, Des Moines, IA, USA). ^§^ Fat thickness over the center of the LM between the 12th and 13th ribs. * Carcass yield was estimated as (Fat thickness × 0.10) + 0.40.

**Table 4 animals-13-02547-t004:** Effect of treatments on whole cuts of finishing lambs.

	Treatments					
Item	MON	MON + BS	MON + BS + EO	BS	SEM	*p*-Value
Whole cuts (as a percentage of CCW)						
Forequarter	46.75	45.69	45.98	45.93	0.387	0.29
Hindquarter	42.37	42.07	43.60	42.69	0.643	0.40
Neck	9.52	9.06	9.90	9.09	0.440	0.51
Shoulder IMPS206	8.84	9.61	8.45	8.97	0.542	0.54
Shoulder IMPS207	17.01	15.90	16.73	16.84	0.464	0.38
Rack IMPS204	7.59	7.53	7.65	7.58	0.191	0.97
Breast IMPS209	5.07	4.58	4.45	4.60	0.254	0.37
Ribs IMPS209A	8.11	7.79	8.00	8.04	0.203	0.71
Loin IMPS231	8.38	8.30	8.57	8.29	0.196	0.74
Flank IMPS232	7.01	6.93	7.11	6.92	0.201	0.90
Leg IMPS233	27.12	26.82	27.48	26.76	0.586	0.69

**Table 5 animals-13-02547-t005:** Effect of treatments on the visceral mass of finishing lambs.

	Treatments ^†^					
Item	MON	MON + BS	MON + BS + EO	BS	SEM	*p*-Value
Empty body weight, % of full weight	91.16	91.91	90.18	90.52	0.530	0.52
Organs, g/kg of empty body weight						
Stomach complex	29.28	29.26	29.03	31.68	1.584	0.61
Intestines	46.11	48.51	43.88	46.39	2.162	0.54
Liver/spleen	14.64	14.42	13.07	13.76	0.878	0.88
Heart/lungs	20.03	20.78	17.69	19.46	1.495	0.51
Kidney	2.79	2.63	2.32	2.61	0.176	0.34
Omental fat	33.95	33.41	32.85	34.70	0.836	0.13
Mesenteric fat	12.87	11.20	12.57	12.65	0.857	0.19
Visceral fat	46.82	44.61	45.42	47.35	0.775	0.75

**^†^** MON = Monensin 28 mg/kg diet DM (Rumensin 90^®^, Elanco Animal Health, Indianapolis, IN); MON + BS = combination of MON plus 2 g/kg diet of a product contained *Bacillus subtilis* 2.2 × 10^8^ CFU (CLOSTAT dry, Kemin Industries, Des Moines, IA, USA); MON + BS + EO = combination of MON + BS plus 300 mg essential oils (PrintArome, NOREL Nutritición Animal, Queretaro, Mexico); BS = 2 g/kg diet of a product contained *Bacillus subtilis* 2.2 × 10^8^ CFU (CLOSTAT dry, Kemin Industries, Des Moines, IA, USA).

## Data Availability

The data supporting this study’s findings are available from the corresponding author upon reasonable request.

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
