# Peer review of "Effect of Combining the Ionophore Monensin with Natural Antimicrobials Supplemented in the Last Phase of Finishing of Lambs: Growth Performance, Dietary Energetics, and Carcass Characteristics"

_animals, 2023, doi:10.3390/ani13162547_

Round 1

Reviewer 1 Report

This study evaluated the use of monensin with natural antimicrobials during the late-finishing period. The methods used and replication are adequate for this type of research. 

Line by Line comments are below:

L 18: The ionophore

L 115: replicates and m2

L 139: provided 

L 142: final--> at the conclusion of the experiment

L 186: how was NEg (Mcal/kg) derived from NEm (Mcal/kg) assuming NEg = 0.877NEm-0.41?

L 233: plenty --> completely and solubles to soluble

L 247: ambient heat load

L 252: italicize bacillus subtilis here and throughout 

L 252: suggest to re-word the little tested to has not been investigated thoroughly in lambs?

L 262: similar to Gubbels et al 2023 in TAS who evaluated long-term supplementation in cattle. 

L 285: remove period after utilization 

No major issues detected.

Author Response

Response to REVIEWER 

AU: We are grateful to reviewers for the time and effort in helping improve the quality of the manuscript. The observations were wise and timely which permit the improvement substantially the manuscript. We have addressed the concerns in our revised manuscript accordingly.

All changes and correction made are highlighted in yellow in the corrected version of the manuscript.

Responses

RW: L18: The ionophore

AU: Correction was done

RW: L115: replicates and m2

AU: Correction was done

RW: L139: provided 

AU: AU: Correction was done

RW: L142: final--> at the conclusion of the experiment

AU: Sentence was changed as suggested

RW: L186: how was NEg (Mcal/kg) derived from NEm (Mcal/kg) assuming NEg = 0.877NEm-0.41?

AU: Thanks! The specification of NEg value derivation was made

RW: L233: plenty --> completely and solubles to soluble

AU: Corrections were done as suggested

RW: L247: ambient heat load

AU: Sentence was changed as suggested

RW: L252: italicize bacillus subtilis here and throughout 

AU: Done

RW: L252: suggest to re-word the little tested to has not been investigated thoroughly in lambs?

AU: The sentence was rewording as suggested

RW: L262: similar to Gubbels et al 2023 in TAS who evaluated long-term supplementation in cattle. 

AU: Thanks! The reference was included to strengthen the support of our findings

RW: L285: remove period after utilization 

AU: Done

Reviewer 2 Report

Comments to the manuscript ID 2438333 “Effect of Combining the Ionophore Monensin with Natural Antimicrobials Supplemented in the Last Phase of Finishing of Lambs: Growth Performance, Dietary Energetics and Carcass Characteristics”. This study evaluated the effect of combination of monensin with some additives on the productive efficiency (performance and carcass) during the last phase of lamb fattening.

The room for improvement is always there and I have suggested some major revisions.

Author Response

Response to REVIEWER 2

AU: We are grateful to reviewers for the time and effort in helping improve the quality of the manuscript. The observations were wise and timely which permit the improvement substantially the manuscript. We have addressed the concerns in our revised manuscript accordingly.

All changes and correction made are highlighted in yellow in the corrected version of the manuscript.

RW: Comments to the manuscript ID 2438333 Effect of Combining the Ionophore Monensin with Natural Antimicrobials Supplemented in the Last Phase of Finishing of Lambs: Growth Performance, Dietary Energetics and Carcass Characteristics. This study evaluated the effect of combination of monensin with some additives on the productive efficiency (performance and carcass) during the last phase of lamb fattening.

The room for improvement is always there and I have suggested some major revisions.

Responses

RW: In table 1, the Essential oil in the last treatment (BS) be marked with +++ according with line 124-125.

AU: Thanks! The correction was done. Only BS be marked because it was included alone in these treatment.

RW: Line 121-125. Is not clear why there is not a control diet without additive supplementation, maybe this is the reason why statistical difference where not found, according to the p-value (table 2, 3, 4 and 5).

AU: The main objective was comparing the combination of MON+BS.  Monensin was used as a positive control. Monensin is probably the most studied additive from the decade of 70’s up to present. Based on knowledge of monensin responses and using energy derivation for energetic estimations is a valid approach to make comparisons without Control group.  There are numerous scientific reports directly comparing monensin (positive control) with other additives without an un-supplemented control group (i.e., Meschiatti et al., 2016. J. Anim.Sci., 94: suppl 1-659; Heker et al., 2018. Sem Cs Agr 39:261; Meschiatti et al., 2019. J. Anim. Sci.,97:456; Gouvea et al., 2019. J. Agr Sci. 1-13; Araujo et al. 2019., Appl Anim. Sci. 35:117).

RW: Line 166. Based on tabular energy

AU: Correction was made

Line 218. Authors mention that the data was analyzed with a randomized complete block design, of which the initial weight was the blocking criterion. However, results presented in table 2, in specific Initial weight did not showed statistical differences between treatments, so, there is no reason for blocking.

AU: Initial weight is an important factor that affects growth performance and energy utilization of the diet. In the current study, we had differences on the initial weight, thus to avoid contaminating effects of initial weight, the use of the analyses of covariance or blocking at start the experiment is recommended. We prefer blocking. The purpose of grouping experimental units (blocking) is to have the units in a block as uniform as possible. Each treatment can have 2 or more blocks (in the current experiment we formed 4 blocks). The basic principle of block design is that blocks must be very similar within blocks, but very different among blocks. Thus, the average value (in this case, initial weight) of all blocks result very similar between treatments. This is the reason why the average weight of each treatment was very similar (non-statistical different) due to blocking. Thus, the main objective of blocking was fully accomplished (reduce variation in of an independent secondary variable, in this case, initial weight)

RW: Line 228, Authors point out that the daily average net intake of MON was 37 mg/ lamb. However, that could happened in treatment BS (1.33 kg of DMI, value taken from Table 2) without MON.

AU: Good point!! We recalculate the intakes as 32 mg, 2.4 g and 341 mg for MON, BS and EO, respectively. The data was corrected in the text. Thanks!

RW: In table 2 and according to the line 241. MON decreased DMI in contrast with the BS treatment, why the p-value is higher than 0.05?

AU: Oups! This was a mistake; the proper value is 0.04. Thanks!

RW: Line 236-238. How did you get the CHO to NDF ratio value 4.3, if you did not present CHO values.

AU: You right. The sentence was rewording as follows: Although solubles CHO was not determined in diets, according NRC [19] the estimated solubles CHO in diet was 64%, representing soluble CHO to NDF ratio of 4.3.

RW: Line 248. Lambs received BS alone, showed 19.0% and 14.3% greater ( ) average ADG and dry matter intake (DMI) than lambs received MON+BS+EO not correspond to the p-values (0.13 and 0.11). To me, the text is wrong due to the statistical differences were not found.

AU: The text is right. The p-value shown in the column is the general p-value of treatments. When separate contrast by mean treatments, then BS vs MON+BS+EO had greater ADG (p=0.04) and DMI (p=0.02). Editor insisted that the general p-value of treatments be shown in a column. This can result in confusion in cases where the general p-value of treatment is not significant, but in the case of a specific contrast if it is. In order to clarify, the specific p-value of contrast of each was inserted in text

RW: Line 279, What MP means?

AU: Microbial protein. This was specified as suggested

RW: Line 281. Thus, improves on dietary energy in lambs that received MON+BS could be by BS promotes greater microbial protein

AU: Done

RW: Line 301. Lactobacillus acidophilus plus S. cerevisiae scientific names must be written in italics. Also, if a scientific name has not been mentioned, it shouldn´t be abbreviated.

AU: This observation was corrected here and through the document

RW: Inconsistences to call the treatments were found in some tables as follow: Item MON MON+BS MON+BS+EO BS Item Control MON EO+D3 EO+BS

AU: Thanks! This error was corrected in Tables 4 and 5.

RW: Conclusions I consider to rewrite this section in the way that the treatments were designed. For example, MON and combinations were not effective as BS was. Also, point out that combinations can have an antagonist effect. The warning about the interpretation of your results is not necessary.

AU: Your observation and suggestion was incorporated in conclusion. Warning was removed as suggested.

Reviewer 3 Report

The manuscript investigates the effects of Monensin (MON) and Bacillus subtilis (BS) on the growth performance, energy efficiency, and carcass characteristics of fattening sheep. The study of the combined use of MON and probiotics like BS is significant for animal husbandry since Monensin is still allowed in feed in some countries. The experimental methods are described in detail, and the references cited are relatively abundant. However, there is no control group designed without the addition of both MON and BS, making it difficult to demonstrate the individual effects of MON and the combined effects of MON+BS. The experimental group can only be compared with the BS group, but the results still do not clarify the application effects of MON+BS. Only the energy efficiency is superior to the BS group, but the energy efficiency is based on estimated values, not measured values.   Specific Comments:   Bacterial species names need to be italicized.   In Table 1, in the composition of the premix, there are repetitions of phosphorus and phosphate rock, as well as magnesium and MgO. It is suggested to modify the composition of the premix.   Line 208: "The difference between full and washed digesta-free GIT was subtracted from the SBW to determine empty body weight (EBW)." Why is SBW used here? In my opinion using BW would be better.   Line 213: "including digesta," and line 214: "digesta-free." These statements appear somewhat contradictory.   Line 204: The composition of the digestive tract should not include the liver, pancreas, and gallbladder. It is recommended to modify the description accordingly.   Line 236: The sentence is somewhat difficult to understand. What does "a ratio soluble CHO to NDF ratio of 4.3" mean? It is suggested to be more specific.   In the discussion section, both the first and third paragraphs discussed DMI and NE, the main ideas of each paragraph are not very clear. It is recommended to revise them. The speculation in the third paragraph about factors affecting energy efficiency lacks experimental data support, and personally, I think that increasing propionic acid concentration and microbial protein synthesis are not the primary influencing factors for energy efficiency.

Author Response

Response to REVIEWER 3

AU: We are grateful to reviewers for the time and effort in helping improve the quality of the manuscript. The observations were wise and timely which permit the improvement substantially the manuscript. We have addressed the concerns in our revised manuscript accordingly.

All changes and correction made are highlighted in yellow in the corrected version of the manuscript.

Reviewer 3

RW: The manuscript investigates the effects of Monensin (MON) and Bacillus subtilis (BS) on the growth performance, energy efficiency, and carcass characteristics of fattening sheep. The study of the combined use of MON and probiotics like BS is significant for animal husbandry since Monensin is still allowed in feed in some countries. The experimental methods are described in detail, and the references cited are relatively abundant. However, there is no control group designed without the addition of both MON and BS, making it difficult to demonstrate the individual effects of MON and the combined effects of MON+BS. The experimental group can only be compared with the BS group, but the results still do not clarify the application effects of MON+BS. Only the energy efficiency is superior to the BS group, but the energy efficiency is based on estimated values, not measured values.

AU: The main objective was comparing the combination of MON+BS.  Monensin was used as a positive control. Monensin is probably the most studied additive from the decade of 70’s up to present. Based on knowledge of monensin responses and using energy derivation for energetic estimations is a valid approach to make comparisons without Control group.  There are numerous scientific reports directly comparing monensin (positive control) with other additives without an un-supplemented control group (i.e., Meschiatti et al., 2016. J. Anim.Sci., 94: suppl 1-659; Heker et al., 2018. Sem Cs Agr 39:261; Meschiatti et al., 2019. J. Anim. Sci.,97:456; Gouvea et al., 2019. J. Agr Sci. 1-13; Araujo et al. 2019., Appl Anim. Sci. 35:117).

RW: Specific Comments:   Bacterial species names need to be italicized.  

AU: This observation was corrected through the document.

RW: In Table 1, in the composition of the premix, there are repetitions of phosphorus and phosphate rock, as well as magnesium and MgO. It is suggested to modify the composition of the premix.  

AU: Correction was made following your suggestion

RW: Line 208: "The difference between full and washed digesta-free GIT was subtracted from the SBW to determine empty body weight (EBW)." Why is SBW used here? In my opinion using BW would be better.

AU: You right. The text was wrong exposed. We use the fasted final body weight. The statement was corrected as follows “digesta-free GIT was subtracted from the final BW to determine empty body weight (EBW) 

RW: Line 213: "including digesta," and line 214: "digesta-free." These statements appear somewhat contradictory.  

AU: There are no contradictions. The full visceral mass was calculated by the summation of all visceral components (stomach complex, liver, heart, lungs, etc.) including digesta. While the stomach complex was calculated as the digesta-free sum of the weights of the rumen, reticulum, omasum and abomasum. However, the variable “full visceral mass" is not presented in Table, then, in order to avoid confusing the concept “full visceral mass was removed from the text.

RW: Line 204: The composition of the digestive tract should not include the liver, pancreas, and gallbladder. It is recommended to modify the description accordingly.  

AU: Agree! Description was modified as: Components of the digestive tract (GIT), including stomach complex (rumen, reticulum, omasum, and abomasum), liver, small intestine (duodenum, jejunum, and ileum), and large intestine (caecum, colon, and rectum) were removed and weighed

RW: Line 236: The sentence is somewhat difficult to understand.

AU: In order to improve clarity, the statement was reworded as follows:  The basis of this is not completely understood, but high soluble carbohydrates -to-NDF ratio in diet (i.e. >4), and environmental factors (high-ambient heat load), could be the main factors that can affect the magnitude to MON response regarding to the improves on efficiency of utilization of diet energy [32,33]. Because the present experiment was performed under favorable environmental conditions, then proportion CHOS: NDF in diet could be the factor that explains the low increase (1%) of the observed over expected dietary NE for lambs receiving MON. Although solubles CHO was not determined in diets, according NRC [19] the estimated solubles CHO in diet was 64%, representing a high soluble CHO to NDF ratio of 4.3.

RW: What does "a ratio soluble CHO to NDF ratio of 4.3" mean? It is suggested to be more specific.

AU: Sentence was rewording to improve clarity  

RW: In the discussion section, both the first and third paragraphs discussed DMI and NE, the main ideas of each paragraph are not very clear. It is recommended to revise them.

AU: Paragraphs was revised and improved when necessary

RW: The speculation in the third paragraph about factors affecting energy efficiency lacks experimental data support. Personally, I think that increasing propionic acid concentration and microbial protein synthesis are not the primary influencing factors for energy efficiency.

AU: Changes on dietary energy utilization efficiency can be affected by several intrinsic and extrinsic factors. The main explanations of the increased efficiency on diet energy utilization by additives supplementation (such antibiotics and probiotics) are changes on ruminal fermentation, increases on amino acids to intestine and increases on intestinal nutrient absorption, and finally by improves on cellular metabolism (mainly by reduction of cellular oxidative stress). Following your suggestion discussion was improved with inclusion of reports that mention the effect of probiotics on cellular oxidative stress.

Round 2

Reviewer 2 Report

I have reviewed the manuscript, authors have answered and corrected all my questions and observations properly. To me is fine to continue with the publication process. 

Thank you for taking into account my comments

Author Response

Thank you for your invaluable support in reviewing this manuscript.

Kind regards!

Alejandro Plascencia

Reviewer 3 Report

The author's response to the previous review comments is commendable, as they have effectively addressed the concerns raised. Additionally, I still want to offer a suggestion. In the discussion section pertaining to the influencing factors of energy efficiency, I recommend placing a stronger emphasis on the role of microflora and the fermentation process, rather than focusing on the microbial protein content entering the small intestine. It is important to recognize the distinction between protein utilization efficiency and energy use efficiency. I encourage the authors to consider referencing the paper titled "Specific microbiome-dependent mechanisms underlie the energy harvest efficiency of ruminants. ISME J. 2016, 10, 2958-2972" to support and enrich this aspect of the discussion. Furthermore, there are still species names not written in italics; it is recommended to verify and make necessary corrections.

Author Response

AU: We are grateful to reviewers for the time and effort in helping improve the quality of the manuscript. The observations were wise and timely which permit the improvement substantially the manuscript. We have addressed the concerns in our revised manuscript accordingly.

All changes and correction made are highlighted in yellow in the corrected version of the manuscript.

Reviewer 3 Comment & Responses

RW: The author's response to the previous review comments is commendable, as they have effectively addressed the concerns raised. Additionally, I still want to offer a suggestion. In the discussion section pertaining to the influencing factors of energy efficiency, I recommend placing a stronger emphasis on the role of microflora and the fermentation process, rather than focusing on the microbial protein content entering the small intestine. It is important to recognize the distinction between protein utilization efficiency and energy use efficiency. I encourage the authors to consider referencing the paper titled "Specific microbiome-dependent mechanisms underlie the energy harvest efficiency of ruminants. ISME J. 2016, 10, 2958-2972" to support and enrich this aspect of the discussion. Furthermore, there are still species names not written in italics; it is recommended to verify and make necessary corrections.

AU: Honorable reviewer, the paper "Specific microbiome-dependent mechanisms underlie the energy harvest efficiency of ruminants. ISME J. 2016, 10, 2958-2972” has already been referenced in the original manuscript (as reference #45). In the original manuscript (and totally agree with your observation), we mentioned that the main cause of changes on energy efficiency is the change in ruminal fermentation patterns (L273-282). However, other causes can assist the increase on energy retention as well, within these, increases in MN flow, increases on intestinal absorption rate, and by better capacity to cellular antioxidant mechanisms, which are mentioned in the manuscript as well.  Anyway, following your wise suggestion we rewrite the statement as: As mentioned previously, probiotics favors changes on ruminal microorganisms and gastrointestinal health analogous to antibiotics. An increase in the ability to obtain energy from feeds, improving the energy efficiency, is explained by the shift to a ruminal microbiome that is less complex, but more specialized to support the host's energy needs [45]. Those authors, indicate that increases on ruminal propionate with decreases on ruminal acetate and methane production, as happens when probiotic or ionophore are supplemented, is the one of the main explanations of the increased efficiency on diet energy utilization.

We recognize the distinction between protein utilization efficiency and energy use efficiency. But it is important to note that several reports have determined that increase in duodenal microbial flow can impact on the efficiency of energy utilization (Zinn and Shen, 1998; May et al., 2014; Castillo-López et al., 2019, among others). For this cause, we mentioned it.

RW: Furthermore, there are still species names not written in italics; it is recommended to verify and make necessary corrections

AU: We carefully checked the manuscript and now all species names are in italics